# Impact of Meloxicam Administration in Cows Prior to Caesarean Section on the Efficacy of Passive Immunity Transfer in Calves

**DOI:** 10.3390/ani13010037

**Published:** 2022-12-22

**Authors:** Raphaël Guatteo, Caroline Lesort, Gwenola Touzot-Jourde

**Affiliations:** 1INRAE, Oniris, BIOEPAR, 44307 Nantes, France; 2Surgery-Anesthesia Unit, Oniris, 44300 Nantes, France

**Keywords:** cattle, caesarean section, analgesia, meloxicam, passive immune transfer

## Abstract

**Simple Summary:**

The improvement of the welfare of cattle is crucial. The aim of this study was to assess the putative benefit of adding pre-emptive analgesia (namely meloxicam) to basic anesthesia during caesarean section (c-section) in beef cows to the efficacy of colostrum intake by the newborn under the assumption that this latter intake could be improved if the dam is not pained. The results demonstrate for the first time the benefit of meloxicam priori to c-section to improve the efficacy of immune passive transfer (an increase of about 3 g/L of IgG in the blood sera of calf coming from dam receiving meloxicam). This is an additional argument to convince farmers and veterinarians to use NSAIDs.

**Abstract:**

The objective of this study was to assess in the Charolais cow–calf system, the benefit of meloxicam administered to cows prior to c-section to the efficacy of passive immune transfer to the newborn under the assumption that providing analgesia to the dam could lead to an earlier or longer colostrum intake. This study was performed in Burgundy, France in two veterinary private practices. Colostrum quality, delay between the end of the c-section and the first spontaneous colostrum suckling, and the 24 h after birth calf serum IgG content from cows treated 15 min prior to c-section with meloxicam subcutaneously (0.5 mg/kg) (n = 22) or without analgesia (n = 26) were compared. No significant differences were observed in the quality of the colostrum nor the delay between the end of the surgery and the first spontaneous colostrum suckling between treatment groups. However, the number of calves showing a better transfer of passive immunity (IgG content >15 g/L) was significantly higher (*p* = 0.023) among those originating from dams receiving meloxicam prior to c-section. This effect was notably observed in multiparous cows (*p* = 0.041). This study confirms that pre-emptive analgesia in cows prior to c-section benefits the calf through an improved colostrum intake that is of paramount importance for calf short- and long-term survival.

## 1. Introduction

Over the past decade interest in farm animal welfare and pain management has grown substantially. However, despite the recognition that several conditions are painful in cattle (e.g., mastitis, lameness, dystocia and caesarean section), providing analgesia is not always a common practice [1,2]. A recent survey conducted in the pasture dairy-based system revealed that while 76% of veterinarians declared frequently use of Non-Steroidal Anti-Inflammatory Drugs (NSAIDs) when performing a c-section, 98% of farmers would like to see their cows receiving NSAIDs. This restricted usage of analgesia, mainly NSAIDs, can be assumed to be due to the lack of scientific and obvious evidence of the benefits of such treatment including economic aspects [3]. Meloxicam is a nonsteroidal anti-inflammatory drug (NSAID) from the oxicam class with a mean half-life around 26 h. Meloxicam acts through the inhibition of prostaglandin synthesis, and it has anti-inflammatory, analgesic and anti-pyretic properties. Meloxicam is supposed to be a greater COX-2 inhibitor in cattle than other NSAIDs. In recent years, regarding pain management meloxicam was the most studied NSAID, especially in relation to calving [1,2,3].

The literature suggests that pain management related to calving, especially following dystocia, should be further investigated [4]. However, in such experiments, investigation is focused on evaluating the benefit of providing analgesia to cows or in calves following dystocia (rather than the impact on the calf). Some authors suggest that in case of clinical signs of reduced vitality, it is plausible that providing NSAID to calves could improve the time to standing and increase colostrum intake, which would therefore decrease the risk of failure of passive immune transfer [4,5]. Several studies indicate a benefit of providing analgesia to cows or calves suffering after dystocia [6,7,8,9,10]. However, the putative impact of providing analgesia around the c-section on the maternal behavior, especially the allowance of early suckling by the calves, is poorly investigated even though the correct intake of the colostrum by the calf in the first hours of life is crucial for an optimal transfer of passive immunity. Therefore, the aim of our study was to assess the impact of providing meloxicam in Charolaise cows (in a cow–calf system) prior to caesarean section on the efficacy of passive immune transfer in calves, under the assumption that providing analgesia to the dam could lead to an earlier or longer colostrum intake by the calf and finally a better immune transfer.

## 2. Materials and Methods

All procedures were performed in accordance with the European Directive (86/609) and French regulations and conformed to the Guide for the Care and Use of Laboratory Animals (NIH Publication No. 85-23, revised 1996).

### 2.1. Study Sample

The sample size was calculated in order to be able to demonstrate an IgG gain of 2 g/L in the blood of the calf depending on whether or not its mother had received meloxicam. For an expected natural variability of 20% between individuals and an error of 5%, 80 calves were required. For logistic reasons, we were able to recruit 48 cow–calf pairs. 

This randomized blind control trial involved Charolaise cows (n = 48) in a cow–calf system that were subjected to a c-section by veterinarians from two practices located in Burgundy (France). This cow–calf system implied that after birth the calf had to suckle its dam by itself, while in the dairy system colostrum intake can be managed by the farmer. Study enrollment took place between January and March 2014. The veterinary practitioners were called to breeding farms as part of their routine practice. The beef cows that needed to undergo a non-elective c-section were enrolled in this study with the following prerequisites: they were in good general health (stand out, no specific disease or hyperthermia, no anorexia and not depressed), presenting a live singleton calf not deliverable vaginally, the cause of dystocia being not uterine torsion nor hydrops and there was no disease or injury that was likely to cause additional pain. To avoid any variability of the colostrum quality due to breed, only Charolaise cows were included in this study. The cows also had to be housed in free barns to allow free movements of both dam and calf (typical for cow–calf systems in the area). Lastly, to avoid bias in the measurement of passive immune transfer, famers were not allowed to milk the cows or force the colostrum intake/suckling onto the calves. The main assumption of our study was that providing analgesia to the dam prior to performing a c-section would lead to a better or earlier acceptance of the calf nursing and then would allow a longer first suckling. Therefore, any other situation leading to interference with pain occurrence (other diseases concomitant to caesarean section, such as lameness, pneumonia) or maternal behavior (such as sedation with xylazine to restrain the animal) led to the exclusion of the animal from this study. Similarly, in case of twins, the cow was excluded to avoid interference on the interpretation of passive immune transfer at the calf level. Once a cow was identified as fulfilling the inclusion criteria, the veterinarian randomly assigned each cow to either treatment A (Metacam^®^ 20 mg/mL solution for injection (Boehringer Ingelheim Animal Health, Ingelheim, Germany), 0.5 mg/kg bodyweight subcutaneously in the neck 15 min before the beginning of the surgery) or treatment B (no analgesia prior to c-section) taking parity into account (primiparous vs. multiparous, as the colostrum quality is known to increase along with the age of the animal) [11,12]. The investigator of this study arrived to the herd during the c-section following the call from the practitioner and took the samples and analyzed the results by group without knowing the treatment assignment. Each c-section was performed in the farm calving facility on standing cows via a left flank approach. Briefly, local anesthesia was administered using a line block with 80–120 mL of lidocaine hydrochloride 2%. Uterine incisions were adapted to calf presentation and were large enough to prevent uterine tearing on calf extraction. After closure of the skin, aluminum spray was applied to the wound and a systemic antibiotic treatment (13.1 mg of benzylpenicillin and 16.4 mg of dihydrostreptomycin per kg, twice 3 days apart, via intra-muscular injection) was given.

### 2.2. Data Collection and Laboratory Analysis

At the end of the c-section, a composite colostrum (8 mL) sample coming from the four quarters (2 mL each) was collected by the investigator. This colostrum sample was analyzed on the farm using the same brix digital refractometer (Milwaukee model MA 887, Fotronic Corporation 99 Washington Street, Melrose, MA, USA). The results were expressed in Brix value. Twenty-four hours after the surgery, the investigator came back to the farm. A blood sample by jugular venipuncture was collected from the calf and the delay observed by the farmer between the end of the surgery and the calf first suckling was registered by the investigator (less than 2 h, between 2 and 6 h, more than 6 h). The sera of the blood samples were isolated and frozen on the sampling day until subsequent analysis of IgG concentration. At the end of the study period, all the sera were tested by radial immunodiffusion (RID) assay (IDRing Box BOV IgG1, BIGGtest, ID Biotech, 63500 Issoire, France), at the Laboratoire Départemental Vétérinaire de Saône et Loire, to assess the IgG content. The results were quantified and expressed in g/L for values above 8 g/L. Below this threshold the quantification was not considered as reliable enough by the manufacturer and the results were therefore expressed as <8 g/L.

### 2.3. Data Analysis

Statistical analysis started by checking for similarity of the two treatment groups (number of animals per treatment group, parity, rank of caesarean section, surgery duration and colostrum quality) using a Student’s *t* or Mann–Whitney U test to ensure that any variation in the IgG calf sera content could be attributed to the modality of colostrum intake (delayed or prolonged suckling) and not to colostrum quality. In a second step we compared (i) the delay between the end of the c-section and the first suckling observed by the farmer using a Fisher’s test and (ii) the difference in terms of calf IgG serum content according to their dam treatment group. For the latter, the analysis was conducted quantitatively (using a Mann–Whitney U test) and qualitatively to compare the distribution of calves with IgG sera content ≥ 15 g/L (considered as a good passive immune transfer) according to their dam treatment group using a chi-square test. The likelihood of a calf having more or less than 15 g/L of IgG was finally assessed using logistic regression, considering treatment, parity, quality of colostrum and delay between birth and first suckling as putative explanatory variables. The putative side effects of the different treatments administrated to the dams were monitored during the following 7 days.

## 3. Results

The description of the study sample is displayed in Table 1. Statistical analysis showed that the two treatment groups were similar in terms of cow and c-section characteristics. Notably, the colostrum quality assessed using a Brix refractometer was similar between the two groups. The delay between the end of the surgery and the first spontaneous colostrum suckling by the calf was also similar regardless of the treatment group (Table 2). Among the 48 calf sera tested using RID, 18 (eight in the treated group and 10 in the non-treated group) were considered as containing less than 8 g/L. A tendency was observed for a higher mean IgG content in the calf sera from the dams receiving analgesia prior to caesarean section compared to those not receiving meloxicam (18.3 vs. 15.2 g/L) (*p* = 0.061) (Table 3). When considering 15 g/L as the threshold for an efficient and good transfer of passive immunity, there was a significant difference (*p* = 0.023) in favor of the calves from the NSAID treated cows regardless of the parity (Table 4). When considering the parity, the difference was significant only in multiparous cows (*p* = 0.041). No side effect was reported in the two treatment groups during the seven days following the surgery, especially no retained placenta. Regarding logistic regression after univariate analysis, only two variables were kept for the final model (*p* < 0.2) (treatment and delay between birth and first intake). These two variables were significantly associated in the final model with the level of IgG in the calf sera.

## 4. Discussion

In our study, the administration of meloxicam subcutaneously 15 min prior to c-section in Charolaise cows in a cow–calf system was associated with an improved transfer of passive immunity to their calves. To our knowledge, this is the first study investigating a putative benefit of meloxicam to calves when administrated to the cows. Treatment with NSAIDs in cows around dystocia or c-section has already been shown to improve cow welfare, feeding time and milk yield [6,7,8], as well as in calves following dystocia or c-section [9,10,13,14,15]. When given to calves suffering from dystocia or low vitality it can make sense to include NSAIDs in the therapy. Here the originality was to focus on giving meloxicam to the cows before c-section and not waiting for putative negative impact on calves. Despite a lower sample size than expected (48 vs. 80) our findings demonstrate that an indirect positive impact of providing analgesia to cows can be expected on their own calves. This can be an additional argument to convince farmers and/or veterinary practitioners to routinely and systematically use NSAIDs in such situations [2,3,16].

To ensure a good transfer of passive immunity, the early ingestion of an appropriate amount of good-quality colostrum is critical. Indeed, the essence of good colostrum management is the 5Q rules: Quality, Quantity, Quickly, Quite clean and Quantifying [17]. Our results (Table 5) confirm the crucial role of the delay between birth and first intake. In our experiment it was not expected that the treatment could improve the quality of the colostrum. The similarity of the colostrum quality between groups was confirmed using the Brix refractometer, which is considered as a reliable tool for colostrum quality assessment [17]. As a consequence, putative increase in IgG content observed in the calf sera could be attributed either to an early and/or a higher colostrum intake. 

As the exact quantity of colostrum suckled by the calves is impossible to monitor in suckling calves, we chose to describe the delay between the end of the surgery and the first calf spontaneous suckling. This delay was similar between treatment groups. An explanation could be that this delay is more linked to the calf and its vitality [5,7]. In the experiment setting, it was not possible to record exactly the duration of each suckling period. The higher level of IgG observed in the calf sera can theoretically be attributed to the colostrum quantity ingested and/or to individual heterogeneity in the gut absorption of IgG. Our main assumption is that providing pre-emptive analgesia on cows allowed longer or more frequent suckling episodes by calves due to less painful behavior in dams [7,8]. Calf vitality could have also contributed to the differences observed. This assessment, as well as the measurements such as the meloxicam serum level peak 4 h after c-section in dams, were not implemented for practical reasons. The transplacental passage of meloxicam during the first minutes of the c-section may have positively affected the calf vitality. In addition, further studies with a larger sample and other NSAIDs would be useful to confirm our findings and extend them to other pain killers (such as morphinic analgesics for instance). At this stage Meloxicam was chosen (i) due to its supposed greater COX2-inhibitor profile leading then to less side effects and (ii) mainly because this is the most studied NSAID in recent years for pain management related to calving [9,10,13,15,17].

The effect was mainly observed in calves from multiparous dams. As the colostrum IgG content is known to be significantly better in multiparous cows [11,12], it is expected that a higher impact on the calf serum IgG content from multiparous cows for a same additional volume of colostrum intake will be observed, allowed by a better analgesia in cows. As colostrum intake by suckling is associated with a higher risk of failure of passive transfer in calves compared with supervised feeding [18], the results of this study show that providing analgesia to cows subjected to a c-section could contribute to a better transfer of passive immunity and therefore ensure better short- and long-term survival of the calves.

The threshold set at 15 g/L was higher than those classically reported in dairy calves (10 to 12 g/L) in consideration that there is failure of passive transfer [19]. However, other studies proposed serum IgG concentrations up to 15 g/L as cut-off point for defining failure of passive transfer [20,21]. More recent papers led us also to be more ambitious in terms of the level of IgG that could be reached in farms [22]. This choice was made to evidence, if existing, a real improvement of the passive immune transfer. Moreover, the classical thresholds are mainly derived from studies on dairy calves while beef calves are exposed during their first week of age to a greater amount of manure and infectious pressure. Nevertheless, we can notice that despite the fact that the colostrum quality was good on average (Brix value > 22%) [23], several calves (18.3 ± 9.9 and 15.2 ± 9.8) had very low IgG content in their sera, indicating that failure of passive transfer can concern also beef calves, especially with caesarean section. Indeed, the pain and discomfort induced by the surgery, not specifically studied here, can interfere with the maternal behavior of the dam [15].

## 5. Conclusions

To our knowledge, this study demonstrates for the first time the indirect benefit of providing meloxicam in addition to anesthesia during c-section to the calf through a likely longer first colostrum intake. These results could contribute to convincing farmers and veterinarians to implement proper analgesia, especially around parturition.

## Figures and Tables

**Table 1 animals-13-00037-t001:** Description (mean and standard deviation) of the study sample according to treatment group.

Variable	Treatment Group A (Meloxicam Prior to C-Section) n = 22	Treatment B (No Analgesia Prior to C-Section) n = 26	*p*-Value
Age (years)	3.7 +/− 1.5	4.3 +/− 1.9	0.318
Parity	1.7 +/− 1.0	2.2 +/− 1.7	0.206
Cesarean rank	1.2 +/− 0.4	1.2 +/− 0.40	0.939
Surgery duration (minutes)	27 +/− 9.5	26 +/− 8.1	0.260
Colostrum quality (% Brix)	26.2 +/− 4.7	26.8 +/− 5.8	0.668

**Table 2 animals-13-00037-t002:** Distribution of the calves according to the delay between the end of the surgery and the first colostrum spontaneous suckling by the calf as observed by the farmer.

Treatment Group	Early Suckling (0–2 h)	Suckling (2–6 h)	Late Suckling (>6 h)	*p*-Value
A (meloxicam prior to c-section)	8	7	7	1
B (no analgesia prior to c-section)	12	8	6	-

**Table 3 animals-13-00037-t003:** Distribution of the quantitative IgG sera calf content (g/L) according to treatment group (n = 30 calves with precise quantification of IgG content).

Treatment Group	Median	Mean	Standard Deviation	*p*-Value
A (meloxicam prior to c-section) (n = 14)	17.5	18.3	9.9	0.061
B (no analgesia prior to c-section) (n = 16)	12	15.2	9.8	-

**Table 4 animals-13-00037-t004:** Distribution of calves with efficient passive immune transfer according to treatment groups (n = 48) and parity.

Treatment Group	IgG Calf Sera Content < 15 g/L	IgG Calf Sera Content ≥ 15 g/L	*p*-Value
All parity			
A (meloxicam prior to c-section)	8	14	0.023
B (no analgesia prior to c-section)	18	8	-
Primiparous only			
A (meloxicam prior to c-section)	6	7	0.402
B (no analgesia prior to c-section)	7	3	-
Multiparous only			
A (meloxicam prior to c-section)	2	7	0.041
B (no analgesia prior to c-section)	11	5	-

**Table 5 animals-13-00037-t005:** Results of the final model for logistic regression investigating the likelihood of having an IgG level above or below 15 g/L in calf sera 24–48 h after birth.

Effect (<15 vs. ≥15 g/L IgG Calf Sera)	Adjusted Mean	Odd Ratio	Confidence Interval 95%
Treatment A (meloxicam) vs. Treatment B (ref: none) (*p* = 0.03)	24.1 vs. 17.8	3.88	1.01–14.96
Delay for 1st suckling <1 h vs. >4 h (ref) (*p* = 0.02)	28.4 vs. 15.2	5.90	1.12–31.15

## Data Availability

Not applicable.

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
