# Peer review of "Impact of Meloxicam Administration in Cows Prior to Caesarean Section on the Efficacy of Passive Immunity Transfer in Calves"

_animals, 2022, doi:10.3390/ani13010037_

Round 1

Reviewer 1 Report

Please,

See attached file

Author Response

Dear Reviewer, thanks for your comments

please find enclosed a letter indicating how we took into account your valubale comments

Best regards

Reviewer 2 Report

General comments

This study mainly aimed to evaluate the effect of Meloxican on the IgG levels of the calves measured at 24h after C-section. The strongest findings were done by the multivariable regressions whereas calves of the group A have 3.9 more changes to present ≥15g/L IgG and 1st suckling < 1h had 5.9 more change with a large 95%CI. This is a field study with some limitations (see specific comments) but, in my opinion, it can serve as a first approach (I suggest to add “…: A first approach.” or similar (e.g., preliminary findings) in the title. The main limitations of the study should be clearly stated in the discussion. Improvements in M&M and discussion sections are suggested to clarify some issues. Moreover, there are a lot of small language errors throughout the manuscript (e.g., L35: “… (e.g., mastitis …”; L38: “…like…”; L44: ); L163: “…NSAIDS…”;L: “…(Table 5)…”; etc., etc. I request a moderate revision.

Specific comments:

L67: “…they were in good general health…” can you provide a definition of general health (stand out, not depressed, etc.)?

L68: “… the cause of dystocia was not pathological…” you mean fetomaternal disproportion and fetal malposition, insufficient cervical dilatation? I.e., causes without affecting the general health status of the dam? also, there is any change to confirm the degree of calves’ depression at C-section time.  If not, please consider to report the limitations of your study during it discussion.

L137: “almost significant (p=0.061)” mean “a tendency for… was observed”. Please correct

L146: Please remove the subheading “3.2 Tables”

L154: Please add the reference in the second row (Table 5).

L169: The threshold of 15g/L IgG is discussed in this paragraph. I suggest to discuss your finding before to state the validity of this threshold (your discussion is firstly about your results).

L178: I suggest to insert the mean and SD IgG value of low group to reinforce and define “several calves “.

L179-181: The pain and discomfort of the dam is not evaluated as variable in this study. I suggest to re-write this part as a new sentence and add a reference.

L198-199: The lack of measurement of calf vitality is a limitation of the present study. Meloxican serum levels peak around 4 hours after SC administration in dams. Have you taken in mind that the putative transplacental passage of meloxicam during the first 25-30 min. can affect (or not) the calf vitality?

L209-212: ?

Author Response

(The authors gave the same response as above.)

Round 2

Reviewer 1 Report

Dear Authors,

The manuscript has significantly improved in its quality. However, I insist:

The topic primiparous/multiparous - Colostrum quality i important enough to use apropriate references (f there is any) to support your sentence, instead of the very general reference number 5. To change numbers on references is annoying, I agree, but the effort will be worth beneficial. Otherwise, delete this sentence.

Comment about line 167 was not answered, did you forget? Please try to complete this info with 1 or 2 lines. It is important that readers worldwide have a good picture of how the production sustem under study works.

Author Response

Dear Reviewer

thanks for your new valuable comments

as you could see  i have added appropriate reference for partiy effect and ask to the initial remark line 167

Regards

Reviewer 2 Report

Dear authors,

thanks for providing this revised version. All comments/suggestions were adressed. In my opinion, this version is able to be publihed in Animals journal. Please correct few typos  (e.g., cow-calf system, our) during proofs.

Author Response

Dear Reviewier

thanks for your remark. Typos were deleted

Regards
